# Recent Trends in Continuum Modeling of Liquid Crystal Networks: A Mini-Review

**DOI:** 10.3390/polym15081904

**Published:** 2023-04-15

**Authors:** Sanghyeon Park, Youngtaek Oh, Jeseung Moon, Hayoung Chung

**Affiliations:** Department of Mechanical Engineering, Ulsan National Institute of Science and Technology, Ulsan 44919, Republic of Korea

**Keywords:** liquid crystal network, phase behavior, phase transition, spontaneous behavior, soft elasticity

## Abstract

This work aims to provide a comprehensive review of the continuum models of the phase behaviors of liquid crystal networks (LCNs), novel materials with various engineering applications thanks to their unique composition of polymer and liquid crystal. Two distinct behaviors are primarily considered: soft elasticity and spontaneous deformation found in the material. First, we revisit these characteristic phase behaviors, followed by an introduction of various constitutive models with diverse techniques and fidelities in describing the phase behaviors. We also present finite element models that predict these behaviors, emphasizing the importance of such models in predicting the material’s behavior. By disseminating various models essential to understanding the underlying physics of the behavior, we hope to help researchers and engineers harness the material’s full potential. Finally, we discuss future research directions necessary to advance our understanding of LCNs further and enable more sophisticated and precise control of their properties. Overall, this review provides a comprehensive understanding of the state-of-the-art techniques and models used to analyze the behavior of LCNs and their potential for various engineering applications.

## 1. Introduction

A liquid crystal network (LCN) is a type of polymeric material that incorporates liquid crystal (LC) molecules into the network. This is achieved through chemical crosslinking between polymer chains and polymer-functionalized LC molecules. The LC molecules, also known as mesogens, are typically non-spherical and can form various meso-scale phases based on the interactions between themselves and their surroundings. The two most common phases that LC molecules form are the isotropic and nematic phases. In the nematic phase, the LC molecules align along a specific direction known as the nematic director. As the orientation of the molecules becomes more uniform, the order of the LC molecules increases. In the isotropic phase, the LC molecules are randomly distributed, and this order is lost. The resulting phase of the LCN depends on the LC phase at which the crosslinking occurs during polymerization. Flexible polymeric chains fill the gaps between the crosslinking sites and the mesogens, connecting the LC molecules that would otherwise flow freely.

Notably, crosslinking confers interesting behaviors to the material. Firstly, the LC phases, characterized by the orientation and distribution of mesogens, which are found at the crosslinking step, are imprinted within the polymeric network. LCN can be categorized as main-chain or side-chain, depending on the location of LC units in the networks. In main-chain architecture, for instance, LC units are incorporated into the polymer backbone, whereas in the side-chain, LC units are attached as pendant groups. The statistical distribution of polymeric networks thus changes according to the LC phase, which is otherwise completely isotropic as assumed in the classical Gaussian network model. Due to incorporation and increased crystallinity, LCNs demonstrate a substantial change in material properties, including an increase in stiffness and anisotropy [1,2]. Similarly, LC behaviors, such as phase transitions and shear flow, are strongly affected by the polymeric constituents in the molecules’ vicinity.

In addition to changes in mechanical properties such as polymeric conformation and fluidity of the mesogens, unique opto-mechanical coupling behaviors, commonly referred to as phase behaviors, are also observed in the hybridized material even after crosslinking. The coupling indicates the intertwined change in mechanical properties (e.g., stress) and the phase transition of the mesogenic constituents (e.g., opacity). To the best of the authors’ knowledge, the majority of reports consider two types of opto-mechanical coupling behaviors: (1) spontaneous deformation and (2) soft elasticity. Although the literature commonly considers either one of the couplings, it is worth noting that recent findings show that these distinctions are not always mutually exclusive [3].

The first coupling behavior is spontaneous deformation, which originates from the fact that a phase shift between LC phases alternates the molecular shape of the polymer, as shown in Figure 1:

The differences in the stress-free configuration induce deformation to accommodate a change in the metric tensors, which is analogous to warpage caused by residual strain. The term “spontaneous” originates from the fact that the stress state within the body does not change when left unconstrained. As shown in Figure 1, the polymer conformation changes its shape from a sphere (Figure 1a) to a prolate (uniaxial ellipsoid) shape (Figure 1b) as the LC phase shifts from isotropic to nematic. Such polymeric conformation at the mesoscale deforms the macroscopic shape.

Soft elasticity is another type of opto-mechanical coupling. When the LCN material is stretched in a direction different from that of the nematic director (i.e., $R_{z}$ in Figure 1), distances between the crosslinking sites change. The rigid LC material rotates to the stretched direction and effectively relieves the stress, as the increase in strain energy is less energetically preferred to the rigid body rotation (i.e., soft mode) of the LC molecules. When the material deforms by a soft mode, the principal direction of the local polymeric conformation continuously changes, and so does the stress-free configuration. Hence, the stress does not increase (i.e., soft elasticity) or increases only marginally (i.e., semi-soft elasticity). Such opto-mechanical coupling is especially salient in lightly crosslinked materials [5] and distinguishes LCN materials from other types of reinforced composites, as both structural and fluid rheological properties are simultaneously expressed. LCN materials have the property of being able to mechanically adjust their optical properties. The polymeric chain can be altered based on the molecular configuration (main vs. side chain). The monograph [5] disseminates the classical approaches to understanding the underlying physics.

The phase behaviors of LCN materials underpin their continued relevance and the structures made from them. These materials have been of interest to researchers from a broad range of disciplines due to the coupling between two dissimilar physics, making LCNs not only theoretically mesmerizing but also promising multifunctional materials. For instance, they can be used to create strong yet lightweight artificial muscles [6,7] and tunable optical devices [8]. Additionally, thanks to reactive LC molecules, non-contact and remotely controlled deformation can be realized without the need for pneumatic or wired driving forces [9,10,11]. For instance, the mesogenic molecules containing the azo-dye (-N=N-) that change to reactive LC molecules exhibit sensitivity to UV irradiation [10].

Such research directions are especially timely due to recent advances in material development, such as reactive mesogens, and precision manufacturing techniques, such as microscale additive manufacturing. These enable the interplay between the rigid LC molecule and flexible polymer chain to be harnessed in more intricate ways. For instance, the complex texture of LC distribution can be fabricated through 3D printing, facilitating the viscoelasticity of the LC-containing fluid [12], modulation through magnetic fields [13], or selective crosslinking [14], leading to nontrivial, often inelastic behaviors [15].

Fully leveraging such potentials of the material requires understanding and predicting phase behaviors as well as the underlying mechanisms. However, understanding the structural behavior of the material is not straightforward due to its multiphysics (e.g., stimuli-responsive structural change) and multiscale (e.g., scale difference in LC orientability between the macroscopic size of the structure) nature. Such modeling is particularly essential in LCN solids since numerous microscopic parameters affect phase transition [16,17], such as polymer chain length and network connectivity, and macroscopic parameters, including boundary conditions and the shape/topology of the structure, determine overall behavior.

In this respect, several numerical methods have been presented to address these problems by providing a predictive window for these interesting behaviors. Diverse aspects of LCNs ranging from the effect of different molecular compositions to structural behavior [18] to the programming of bending curvatures induced by changing macroscopic shapes of structures [19,20] have been investigated. Furthermore, multiscale methods where one or two numerical methods are intertwined have also been proposed [21,22,23]. Interested readers are referred to recent reviews [24,25] on numerical methods and multiscale methods, respectively.

This review primarily aims to disseminate continuum-based models regarding the phase behavior of responsive materials. Although micro- and molecular-level information is lost in these models, as they are included in the constitutive model in an averaged manner, the continuum model is not only beneficial in terms of simplicity but also its applicability in inverse design, as it can be incorporated within the existing design pipeline.

The following sections are organized as follows: First, we introduce the various phase behaviors of LCN and how they can be understood from the perspective of the Verwey– Warner–Terentjev energy (VWT). Second, we discuss material modeling, including constitutive models that present stress–strain relationships or generalized eigenstrain models. Third, we present the structural analysis of LCN-bearing structures, including boundary conditions, complex fields, and structural instability. Finally, we summarize and propose future research directions.

## 2. Nematic Rubber Elasticity and Opto-Mechanical Coupling

The orientation of the polymer chain segments becomes anisotropic due to the existence of the incorporated LC molecules. The long-range symmetry that characterizes LC, i.e., rotational symmetry, is transcribed onto the LCN, making the material highly anisotropic, which distinguishes the LC-bearing material from typical polymeric materials. Based on the assumption of the anisotropic Gaussian distribution, the neo-classical free energy density w [26] of the LCN material is as follows:(1)$$w=\frac{\mu }{2}tr\left[g_{o}F^{T}g^{-1}F+\alpha F\left(I-n_{o}n_{o}\right)F^{T}nn\right]$$
where g and n denote the metric tensor and orientation vector, i.e., the director of LC, respectively, by which the anisotropy of LCNs is defined. The subscript o indicates the reference state of the material determined upon the crosslinking state, e.g., the initial polymeric conformation, and the tensor without the subscript indicates the current state of the material. One may easily notice that the energy density falls into the category of hyperelastic materials, and the invariants found in Equation (1) are a function of the deformation gradient ***F***, metric tensors, and the orientation vector.

The first term of the energy is the ideal part of the LCN energy, which is reduced to the classical rubber elasticity of shear modulus μ when the metric tensors are isotropic with the same step length a, i.e., $g_{o}=g=aI$. The second term accounts for the non-ideal case [27,28], which describes the energy increase whenever the initial and current director do not coincide. A contribution of the non-ideal energy increase to the overall energy is weighted by α, which is typically derived from experiments.

Without loss of generality, we only consider the theory considering prolate LCN, although the theory can be extended to the oblate case straightforwardly. It is also worth noting that the present review considers these two behaviors independently, although these two characteristic behaviors—the soft deformation and the spontaneous deformation—are not always mutually exclusive from one to the other, as noted in [3], indicating that the theory regarding LCN is not complete.

### 2.1. Soft Behavior of the Liquid Crystalline Network

The rotation of the mesogens that are independent of the polymeric network during the transition is often referred to as the soft mode behavior. The neo-classical energy (Equation (1)), containing comprehensive information regarding polymeric change coupled with mechanical deformation, is widely used to address various phase behaviors found in LCNs. Energetically, during the soft deformation, the strain energy increase is minimal, as is clearly demonstrated in the case when the deformation gradient is $F_{s}=g^{\frac{1}{2}}Wg_{0}^{-\frac{1}{2}}\;$, where W is an arbitrary rotation matrix. The energy is invariant to the rigid body rotation of the mesogen, despite the structural deformation $F_{s}$. Detailed energetical descriptions regarding the soft behaviors can be found in the literature, including [26,29].

Ideally, the rotations of the mesogens of each domain are energetically free, and the internal stress does not increase even with the large deformation of the LCN strip. Therefore, the stress–strain curve exhibits a long plateau at the beginning of the stretch that characterizes the soft elasticity. However, the actual soft mode of LC rotation often manifests an increase in the strain energy to some extent, because the original director imprinted on the polymer is preferred, and thus the rotation is penalized. The degree of non-ideality is governed by the parameter α shown in Equation (1). The non-ideal case is often referred to as semi-soft elasticity. A stress increase is observed throughout the deformation, although the modulus during the phase transition is substantially lower than the uniaxial nematic LCN. The semi-softness is considered as a result of the interplay between mechanical strain and mesogen stiffness.

Such distinctive soft and semi-soft elasticity is well described in Figure 2, adapted from Refs. [30,31]. The work of Urayama et al. [30] is especially notable as it demonstrates that the degree of softness is dependent on the fabrication history.

When functionalized mesogens and monomers are crosslinked in the isotropic phase, the mesogens do not exhibit any preferential orientation (the left of Figure 2a), and the resulting LCN becomes polydomain, which is characterized by coexisting local, sub-micro nematic phases of random orientation. Despite the existence of nematic phases, $g_{0}$ remains an identity matrix due to statistical randomness. Upon being stretched uniaxially, microscopic rotations occur, and the orientational order S increases (Figure 2b) until local nematic phases are formed within the body (the right of Figure 2b). Such a transition showcases the optical–mechanical coupling found in LCNs. The transition alternates the opacity of the liquid crystalline material; the polydomain LCN is opaque in the nematic state because of light scattering randomly but becomes transparent after the transition. A driving force of the polydomain–monodomain transition upon stretch can also be energetically described by setting the step-length tensor at the reference condition; the energy is locally minimized by any deformation gradient $F=g^{\frac{1}{2}}W$ and considering micro-rotation of directors n (i.e., quasiconvexification).

On the other hand, when the crosslinking takes place in the deep nematic phase and is tailored to have a uniaxial orientation to the initial director $n_{0}$, using techniques such as two-step crosslinking [32], monodomain LCNs having uniaxially oriented mesogens are incorporated within the polymer. Such manipulation of the mesoscopic symmetries changes the mechanical properties, such as the elasticity tensor, as well as the soft behaviors. When stretched in the direction perpendicular to $n_{0}$, the elongated polymer conformation induces gradual rotation. As shown in the polydomain case, large displacement and minimal (i.e., semi-soft) or no increase (i.e., ideally soft) in stress are found. The optical property of the stretched LCN, however, is saliently different as the stripe domain texture arises at the plateau region, making the LCN opaque [29], while the material is transparent before and after the plateau due to the uniaxially aligned LC molecules. According to Ref. [29], this behavior is attributed to the fact that different local soft modes (i.e., rotation and shear) and the macroscopic combination can realize the overall shift of mesogenic orientation to the loading direction. As shown in Figure 3, these spatially varying microstructures are aligned parallel to the loading direction and laminated in a way that prevents macroscopic shear from accommodating the boundary condition.

### 2.2. Spontaneous Phase Behavior

Spontaneous deformation of the LCN is another type of phase behavior observed in LCN structures. Such behavior is differentiated from the typical elastic or inelastic deformations widely found in polymers in that the structural configurations before and after deformations are both stress-free when the structure is unconstrained. Ordered polymer conformation induced by crosslinked nematic mesogens is again behind the scenes, as it is in soft mode deformation, but this time, the change in order instead of rotation determines the deformation. The anisotropic, prolate polymeric conformation becomes isotropic as the LC phase found in the LCN changes from nematic to isotropic, i.e., order collapse.

The primitive theoretical model of spontaneous behaviors is found in [33], where the spontaneous behavior $F_{m}$ is directly correlated with a change in the polymeric conformation from g to $g_{0}=I$. The tensorial properties $F_{m\;}$ are assumed to be coaxial with g, as evidenced by experiments [5], since the coupling with the microscopic rotation of the LC molecules (i.e., soft mode) is assumed to be negligible. Of course, there is a complex coupling between these two dissimilar mechanisms, as reported in [34], but the model retains its validity at least on a microscopic level.

By minimizing the VWT energy with respect to the uniaxial stretch deformation, the spontaneous deformation λ is obtained as follows:(2)$$F_{m}=diag\left(\lambda^{-\frac{1}{2}},\lambda^{-\frac{1}{2}},\lambda \right),\;\lambda ={\left(\frac{g_{∥}}{g_{⊥}}\right)}^{\frac{1}{3}}$$
$$g=diag\left(g_{∥},g_{⊥},g_{⊥}\right)\;$$
where the amount of spontaneous stretch λ is found to be a function of polymeric shape parameters, parameterized by $g_{\left(*\right)}$. The subscripts ∥ and ⊥ indicate that the property corresponds to the uniaxial direction of the prolate conformation and the one in the perpendicular direction, respectively. Therefore, spontaneous deformation is governed by the anisotropy of the polymeric shape, as shown in Figure 4, which is determined at the moment of the crosslinking; the amount of deformation increases as the order of the LC molecules at the crosslinking step gets higher.

The correlation between optical and mechanical components also explains the experimental results very well, for instance, the change in the molecular compositions of LCNs, including main-chain networks instead of the side-chain LCNs [36]. The reversibility of the deformation, which also distinguishes LCN materials from other smart materials that deform only in one way and cannot recover, can also be understood in view of λ, because the shape parameters $g_{*}$ are determined by LC phases that alternate in response to the changing surroundings. It is also worth noting that the material is assumed to be incompressible in Equation (2), adapted from [33] following the experimental evidence found in lightly crosslinked material [7,33,36], although there are also glassy nematic solids that are compressible [37,38,39]; in such cases, a shrinkage in the perpendicular direction should be additionally taken into account, but the order–strain correlation remains largely intact.

Microscopically uniaxial spontaneous behavior is also a source of diverse macroscopic deformations, as the parameters that are assumed to be uniform in Equation (2), namely nematic director n and change of anisotropy $g_{∥}/g_{⊥}$, can vary both in temporal and spatial scale. For instance, a planar composite LCN structure, in which each laminate constitutes a different type of LCN and hence exhibits dissimilar spontaneous behavior, can deform in the out-of-plane direction, such as bending and twisting by creating an inhomogeneous gradient. On the other hand, the natural decay of the stimuli within the material can be harnessed to create gradients within the material and induce nontrivial (i.e., nonuniaxial) deformation [40].

In the same vein, there are numerous continuum models that are underpinned by LC-elastic coupling, as shown by the VWT energy. For instance, anomalous inflation behaviors of a nematic balloon, e.g., inflation-induced torsion, are described theoretically as shown in [41,42]. Recently, Liang and Li [43] presented a theoretical model of LCN-based metamaterials; by combining a molecular description of the soft elasticity and numerical simulations, the study shows that LCN metamaterials have a compliant response to light-induced bending, which leads to a transition between strain-softening and strain-stiffening with different effective shear moduli. The authors also develop an analytical model to predict the local stretch in a ligament and force in LCN metamaterials, relating the softening and stiffening responses to the geometric and material parameters.

All the above spontaneous behaviors occur mainly due to collapse. The collapse of the liquid crystalline order within LCNs can also be induced by various stimuli, depending on the type of reactive mesogens within the network. One of the most well-studied forms of spontaneous deformation of an LCN is the thermally responsive one, which is caused by the heating or cooling of the material. For instance, the monodomain LCN strip incorporating thermotropic, prolate LC molecules get shortened in response to elevated temperature up to the critical temperature that clears up all crystalline order. The popularity of this material is partly due to the widespread use of thermotropic liquid crystals (LCs), but their ability to smoothly change their molecular orientation in response to temperature changes and the existence of a well-defined theoretical model of describing the phase transition also makes thermotropic LCs a typical material [5]. In addition to temperature, LCNs can also be influenced by other stimuli, such as light and electric fields, resulting in changes to their mechanical and optical properties.

## 3. Constitutive Modeling of LCNs

This section aims to examine the role of constitutive modeling in the numerical analysis of LCN materials, specifically through the use of the finite element method. This type of analysis is crucial for understanding and predicting the structural behavior of LCN-based continua. Unlike energetic models, which have limitations due to assuming specific deformation models, the continuum-based finite element analysis of LCN is versatile and can handle arbitrary conditions imposed on the structure; hence, it offers a more comprehensive description of the material behavior:

First and foremost, the continuum modeling that yields the constitutive model in the general tensorial form can handle multidimensional behaviors; this capability is pivotal, not only because the geometries of LCN-bearing structures are two- or three-dimensional, but also because the rich phase behaviors originate from anisotropy and spatial inhomogeneity. For instance, soft mode deformation is prevented at the edge of a biclamped crosslinked LCN and decays near the boundary in order to maintain mechanical compatibility. Such interplay between soft elasticity and boundary condition is especially important to understand experimental results [29,44]. In this regard, further research is required.

Secondly, incorporation of the multiple physics within the constitutive model is required to model LCN materials’ characteristic phase behaviors stimulated by various types of stimuli. For instance, thermally induced LC order collapse is captured through the addressing of potential phase transitions such as Landau–de Gennes potential energy; modeling such thermally induced spontaneous behavior requires free energy considerations with regard to multiphysics, including Landau–de Gennes potential energy and the VWT energy. The variational approach is typically used to derive the governing equations and constitutive models of a material, but the low-fidelity phenomenological model of stimuli responsiveness is also widely employed as well.

### 3.1. Relaxation of the Neo-Classical Energy

A significant technical challenge in utilizing VWT energy in finite element (FE) analysis is the non-convex nature of the material’s response to deformation gradients. This is evidenced by the presence of the soft mode causing mesogens to rotate independently from macroscopic deformations, which gives rise to complex optical textures due to the inhomogeneous distribution of rotation at macroscopic scales. As a result of nonconvexity, an increase in the energy state of the structure is effectively mitigated, and the same stress state can be achieved through different strain states (i.e., nonconvex). Moreover, the different degrees of soft mode for each material point are also affected by the compatibility condition within the structure. Being distinctive features of the LCN, the energetic nonconvexity, the mechanisms of the soft modes, and their correlations have received a lot of interest in both physical and engineering research communities by analyzing the energetic landscape of the material response for the prescribed mode of deformation as described in Section 2.1.

However, the development of a constitutive model that accurately captures the multi-dimensional behavior of the material is a complex task. As highlighted in the energetic description of the energy [29], the elastic energy represented by Equation (1) is non-convex due to its dependence on microscopic liquid crystal orientations, which are independent internal variables. The energy minima of the VWT free energy vary according to the phase, making it unsuitable for use in its unmodified form within the standard variational approach, which relies on the strict convexity of the free energy.

In order to address such an issue and propose the finite element model of LCNs, Conti et al. [44] utilized the quasiconvexification technique, which is frequently used to model metallic alloys experiencing binary phase transitions [45,46]. Such a mathematical technique originates from Ball [47,48], who employed the idea of the scale separation of macro-(F) and micro-deformation gradients ($F_{i}$) and allowing the macroscopic deformation gradient to be modified in a way such that the affine deformation is superimposed by microscale perturbation ($F=\left(1-\lambda \right)F_{1}+\lambda F_{2})$) to attain convergence in the sense of averages (i.e., weak convergence), as shown in Figure 5:

The quasiconvexified model of the VWT free energy $W_{qc}$, defined as an infimum of the energy considering all possible admissible microscopic states (i.e., internal variables and deformation gradient), is thereby created [50]:(3)$$W_{qc}\left(F\right)=\left\{\begin{array}{cc} \;\;\;\;\;\;\;\;\;\;\;\;\;\;\;\;\;\;0\;\;\;\;\;\;\;\;\;\;\;\;\;\;\;\;\;\; & \left(\mathrm{liquid}\;\mathrm{phase}\right)\\ \;\;\;\;\;\;\;\;\;\;\;\;W\left(F\right)\;\;\;\;\;\;\;\;\;\;\; & \left(\mathrm{solid}\;\mathrm{phase}\right)\\ \lambda_{1}^{2}+2a^{\frac{1}{2}}\lambda_{1}-3a^{\frac{1}{3}}\;\;\;\;\;\; & \left(\mathrm{smectic}\;\mathrm{phase}\right)\\ \;\;\;\;\;\;\;\;\;\;\;\infty \;\;\;\;\;\;\;\;\;\;\;\;\;\;\;\;\;\; & \left(\det F\neq 1\right) \end{array}\right.$$
where $\lambda_{1}$ is the lowest eigenvalue of the deformation gradient F, and a is a degree of stretch. Different phases denote the degree of possible soft modes; the liquid phase allows energy-free rotation, the smectic phase allows rotation only within the plane, and the solid phase indicates a situation where the load is applied parallel to the director, hence preventing director rotation. It is worth noting that a similar phase diagram is obtained through direct energy minimization of the VWT energy [51]. The idea of relaxation is also explored by de Luca et al. [52], where small deformations and large director rotations are considered, leading to the characteristic stretch-induced shear due to the superimposed soft deformation.

Although these theories provide valuable insights into the overall behavior of the system, they can be challenging to compute explicitly and may not fully capture the finer details (i.e., microstructural evolution) because of the ad hoc assumption imposed on the deformation gradient [53]. In that respect, several alternatives have been proposed to address the non-uniqueness of the solution. Fried et al. [54] presents an alternative way to formulate the free energy by decomposing elastic and nematic energy instead of using the molecular–statistical approach. By using a phenomenological description based on the symmetry of the LC molecules, the nematic energy is found to regularize the energy profile and alleviate the non-uniqueness of the solution.

### 3.2. Variational Modeling of the Constitutive Model

Development of the constitutive model using the variational modeling technique, which is also known as the principle of virtual work, is necessary to extend the continuum modeling of LCN structures. Through the variational technique, the equilibrium stress state of a structure is determined by minimizing the potential energy of the system for the given constraints. Modeling is especially required in modeling the structural behavior of LCN structures since dissimilar materials of different governing physics constitute the LCN.

In this regard, various works have developed constitutive models of LCNs, aiming to reveal the relationship between stress and strain and other internal variables such as the microscopic characteristics of mesogens [34,55,56,57,58]. These models appear to differ in their resulting models and governing equations due to differences in assumptions regarding kinematics and types of energy. Nevertheless, behaviors explained by the model are very similar since they concern similar multiphysical natures of the material, namely polymer elasticity, first- and second- order phase transitions, and distortion energy of the LC molecules, and the solution of the model should asymptotically converge to the energetics solutions and experiments described in Section 2.

Early attempts to describe the coupled behavior include the static continuum approach where small deformations and large rotations are considered and incorporated within polynomial forms of energies assumed phenomenologically [59,60]. Anderson et al. [59] developed a continuum model for nematic elastomers by employing the strain energy proposed by Bladon et al. [61] as well as the Ossen–Frank energy. The model is comprehensive, as the total potential energy incorporates not only the strain tensor but also the orientation and the orientation gradient within its functional. Although limited to small strain and large rotation kinematics due to the linearization, the model has been found to be sufficiently descriptive of the various coupled behaviors; Petelin and Čopič [62] demonstrated that the model is capable of reproducing the experiments that manifest the influence of the relaxation rate on the soft modes. A more comprehensive behavior of the LCN is described employing the ideal symmetry of LC mesogens by Lubensky et al. [60], where the elastic energy, nematic–isotropic transition energy (i.e., Landau–de Gennes energy), and the ad hoc coupling energy are considered. However, as noted by [57], these models focus on specific types of soft elasticity and have limitations in describing mechanical behaviors of LCNs, e.g., semi-soft elasticity as shown in [29,49] and an asymmetric stress tensor due to internal coupling.

Jin et al. [57] first proposed a consistent thermomechanical approach in the constitutive modeling of LCNs based on the variational technique. The free energy L is defined by the VWT elastic free energy $L_{el}$, the phase transition energy $L_{LdG}$, and two popular constraints of volume preservation and orthogonality of the directors $e_{i}\;\left(i=1,2,3\right)$:(4)$$L=L_{el}\left(F\right)+L_{LdG}\left(Q\right)+p\left(\det F-1\right)+\xi \left(e_{i}\cdot e_{j}-\delta_{ij}\right)\;$$
where p and ξ indicate the Lagrange multipliers for the constraints. F and Q are the deformation gradient and order tensor parameters, which are state variables of the system. Using the Clausius–Duhem inequality and ignoring the inner couple for the sake of brevity, the variationally consistent constitutive model, as well as the governing equations for the nematic solids, is derived as shown:(5)$$\sigma =-pI+\mu \sum_{i}^{3}g_{i}^{-1}B_{i}\;e_{i}e_{i}-\mu \sum_{i}^{3}g_{i}^{-2}B_{i}\frac{\partial g_{i}}{\partial Q}+2\frac{\partial L_{LdG}}{\partial Q}=0\;$$
where $B=Fg_{0}F^{T}$ is the effective left Cauchy–Green tensor, and Q is a scalar order parameter. Equation (5) indicates mechanical and LC order equilibrium. It is worth noting that the coaxiality between g and Q is assumed to be preserved throughout the deformation due to the absence of the inner couple, hence removing the necessity of the dynamic equation describing the mesogenic directors’ rotation.

The comprehensive behavior of LCNs—phase behaviors and semi-soft elasticity—is analyzed based on Equation (4) for the given boundary condition, normal deformation λ, and shear deformation k with the initial director $n_{0}$ (Figure 6a). The spontaneous deformation describing the relationship between the obtained heat-induced order change is also reproduced in the premise of stress-free or hydrostatic pressure applied to the body and by changing the eigenvalues of Q (Figure 6b). Additionally, the soft behavior of the order–mechanical coupling, i.e., stress–strain condition in two-dimensional LCN material, is also predicted through the model (Figure 6c), where the stress and stretch correlation is not monotonic.

As a result, comprehensive parametric studies can be conducted over a wide range of manufacturing conditions (e.g., initial director) and operating conditions (e.g., temperature) using the model. These solutions not only agree well with the semi-analytic solution of soft behavior found in [29], but also demonstrate the effect of nematic orientations and orders on the stress–strain relation. Moreover, the authors identified the complex relationship between the order parameter of LCNs and their mechanical behavior, revealing the potential of phase behavior modulation using the LC properties (e.g., director field).

The comprehensive thermo-order-mechanical static model (Equation (5)) was later applied to various LCN analysis models thanks to its comprehensiveness and capability to explain multiphysics behavior. For instance, Lin et al. [63] derived a simple constitutive model of the LCN by linearizing the general thermomechanical model. Nevertheless, these models are limited to the static equilibrium of the LCN structure that assumes that the LC order as well as polymeric conformation change is taking place instantaneously, and it is not capable of explaining time-dependent behavior; this is a salient limitation since time-dependence is widely observed in LCN experiments because of the viscoelastic behavior and delayed transition of LC molecules owing to their polymeric constituents.

In this respect, the energy dissipation and dynamics of LC mesogens have been considered in the literature [55,58,64,65]. One notable work regarding the dynamic constitutive model is by Oates and Wang [55]. The authors assume that the LC order change creates entropy within the system, affecting the heat flux following Duhamel’s law. This is important in modeling soft elasticity, as the model considers entropy creation and heat exchange as energy-dissipating mechanisms. The pseudo-director $n^{*}$, which parameterizes the tensor order parameter, is assumed to be a work conjugate to LC stress terms. The model also avoids the problem of a non-convex functional landscape [29,44], since the VWT energy is replaced by a standard hyperelastic model.

The governing equations are thus:
$$\nabla \cdot P+B=\rho_{0}\;\dot{V}$$
(6)∇ξ+γ+ϕ=0
$$\dot{\theta }=\dot{\theta }\left(\nabla \theta ,\nabla^{2}\theta ,n^{*},F\right)$$
where P, B, F and V found in the first equation indicate the first Piola–Kirchoff stress, body force, deformation gradient, and the velocity of the body, respectively. The coupled equation is shown in the second equation where the coupled stress ξ, an order higher than typical mechanical stress, is coupled with the nematic director body force, γ. The dynamic correlation of the temperature θ and the mechanical and liquid crystalline variables are readily solved through finite element software. As a result, the model is found to be capable of simulating both monodomain and polydomain-to-monodomain LC texture evolution under the condition of finite strain assumption.

As noted by the authors, however, the model is largely phenomenological as it does not consider the inner coupling between LCs and polymeric constituents by using a typical neo-Hookean energy model, although the results underpin that the model has reasonable accountability in discussing the complex relationships found in LCNs, e.g., domain structure evolution for the given external loads and the nontrivial contribution of the Frank elasticity. A similar idea has been investigated by Keip and Bhattacharya [66], where the inner coupling is considered and evolution equations of the orientational order (Allen–Cahn type) and nematic orientation (Landau–Lifshitz type) are formulated.

More recently, the dynamic constitutive model considering the inner couple of the LCN was developed by Zhang et al. [58]. In the view of the virtual power principle, the variation in the total working power W and the dissipation function R can be written for a general continuum theory of LCN in isothermal conditions:(7)$$\frac{d}{dt}\left(\delta K+\delta U\right)+\delta R=\delta W$$
where K and U indicate kinetic and internal energy, respectively, and W is energy considering external agents (e.g., external load) and constraints, similar to Jin et al. [57]. Based on the Ericksen–Leslie theory, the non-symmetric component of stress pinpointing the existence of the inner couple is derived. A simple quadratic form of the Rayleigh dissipation model R is used, which greatly simplifies the overall derivation:(8)$$R=\frac{1}{2}\eta_{0}\;tr\left({\dot{\epsilon }}^{2}\right)+\frac{1}{2}\eta_{n}{\left(n^{\nabla }\right)}^{2}$$
where $\eta_{0}$ and $\eta_{n}$ are positive constants representing viscosity with respect to strain rate $\dot{\epsilon }$ and nematic director’s objective rate $n^{\nabla }$, respectively. The constitutive models regarding stress and director for the given state variables are derived as:$$\sigma =\sigma^{b}\left(u,\;\dot{\epsilon },n\right)+\sigma^{EL}\left(\nabla n,\;\dot{n},\;W\right)$$
(9)$$\eta_{n}\dot{n}\times n=\eta_{n}W\left(n\times n\right)-2\tau^{b}-Kn\times \nabla^{2}n-{\hat{G}}^{n}$$$\sigma^{b}$ and $\sigma^{EL}$ indicate the bulk and the LC stress, respectively, where the director–bulk stress coupling is originated by the VWT energy model employed in the work. It is worth noting that the W-$\sigma^{EL}$ coupling and the evolution of the director found in the second equation of Equation (8) are asymptotically reduced to the work of Jin et al. [57] when the dynamic effects are ignored. The dynamic equation regarding mesogenic rotation, i.e., $\dot{n}$**,** is solved via the finite element software COMSOL.

There are several research avenues that have been pioneered based on these constitutive models. For instance, Wang et al. [56] extended the model by incorporating nonlinear viscosity theory, by which the viscous behavior of mesogenic rotation is considered. A constitutive relationship between the rate of deformation tensor of the elastic and viscoelastic parts to the viscous stress is established. In order to do so, the authors first modify a neo-Hookean free energy density to Gent’s model, which is also a hyperelastic strain energy that captures different regimes observed experimentally. The non-equilibrium part of the free energy density, of which material parameters represent viscous behavior originating from network evolution, is then additively superposed to the total energy. This model can simulate the finite instantaneous stress response as well as experimentally observed stress relaxation, especially in lightly crosslinked LCNs. As a result, the material hysteresis as well as the elastic strain stiffening behaviors that represent molecular re-orientation are observed, which are more realistic rate-dependent stress responses. It is also worth noting that the development of advanced models is still under investigation, e.g., the Föppl–von Karman type of constitutive model [67].

### 3.3. Eigenstrain Models

The eigenstrain modeling technique in the field of solid mechanics broadly refers to the idea of modeling changes in the stress state induced by embedded inhomogeneity within the material by internal reciprocal strain. Although it was first devised to represent the effect of inclusions having diverse directions and shapes incorporated within composite materials, the technique was later extended to diverse engineering applications where internal strains evolve, for example, warpage due to residual stress and phase transition leading to pseudo-elasticity [46,68], to name a few. Modeling the spontaneous behavior of smart materials is not an exception, as the expansion or contraction of the material in specific directions in response to an external stimulus can be accounted for by the eigenstrain. Being a surrogate model correlating the deformation with the stimuli imposed upon the structure, the eigenstrain model is a powerful tool for predicting and understanding the mechanical behavior of multifunctional structures. It is also worth noting that the idea of using eigenstrain to model macroscopic deformation should date back long before the emergence of smart materials, as witnessed by the modeling of bending behavior of continua made of bimetal [69].

In this review, we adopt the simplest eigenstrain model to describe the spontaneous behavior by using the following equation:(10)$$\epsilon =\epsilon^{e}+\epsilon^{*}$$
where $\epsilon^{*}$ and $\epsilon^{e}$ indicate the eigenstrain and the elastic strain, respectively. The total strain ϵ is assumed to be additively decomposed into these two terms based on the infinitesimal strain assumption, and it describes widely known phenomena where a free-floating structure changes its shapes without changing the stress state within it. In a one-dimensional problem, the eigenstrain $\epsilon^{*}$ is equivalent to the gradient of stretch λ discussed in Equation (2). Being a generalized model of internal stress, the eigenstrain is anisotropic and is therefore represents typical LCN spontaneous behaviors; microscopic change in the polymeric conformation can be represented by [12,70,71,72,73]:(11)$$\epsilon^{*}=\epsilon_{∥}nn+\epsilon_{⊥}\left(I-nn\right)\;$$
$$\epsilon {\;}_{⊥}=\nu^{*}\;\epsilon_{⊥}\;$$
where strain induced along the nematic orientation n is $\epsilon_{∥}$, while the perpendicular to the director is $\epsilon_{⊥}$ t. It is worth noting that the orientation vector can either be a microscopic [19] or macroscopic [57] description of the LC phases.

Although simple, various real-world experiments regarding LCNs can be addressed by simply tailoring the model or proposing constitutive relations between $\epsilon^{*}$ with stimuli (e.g., heat) and internal variables (e.g., mesogenic director). For instance, $\nu^{*}$, which is the Poisson ratio corresponding to the spontaneous deformation of the polydomain LCN, is zero, while in the monodomain LCN, it has a negative value that describes experimental results [74] originating from the different metric tensors evolved within the body [71]. Additionally, stimuli-dependent spontaneous behaviors, such as the different degrees of LCN’s uniaxial shrinkages depending on the magnitude of heating [11], can be modeled by simply representing $\epsilon^{*}$ as a function of properties related to the stimuli; for instance, $\epsilon_{∥}$ is proportional to the magnitude of the stimuli [40].

The flexibility of the model is widely employed to discuss and provide grounds for various nontrivial spontaneous behaviors, especially the reversible out-of-plane deformation of LCN structures that notably distinguish LCNs from other types of smart materials. These types of behaviors include patterned out-of-plane deformation [37] and the development of anticlastic curvatures [75]. It is worth noting that such behaviors are not well addressed through the energetics of the one-dimensional model (Section 2.2); although the energetic description reveals well regarding the governing mechanisms of the spontaneous deformation, the model is incapable of modeling general behaviors because it assumes homogeneity (e.g., uniform temperature change) and uniaxiality (e.g., same nematic directors).

Nevertheless, one may easily notice that the constitutive model regarding the eigenstrain is typically determined through experiments. The resulting model is inevitably of low fidelity because the complex physics intertwined within the LCN material are observed in an averaged manner and thus lose many important microscopic details such as the types of molecules [76] and stimulus–material interaction [77].

In order to increase the fidelity and predictability of the model, the multiscale/multiphysics nature of the spontaneous behaviors has also been investigated. First of all, the sequential multiscale model is employed in [23] (Figure 7a); by facilitating microscopic information being obtained using small scale simulations, e.g., the full-atom molecular dynamics [18] regarding thermotropic LCN behavior, the eigenstrain is modeled based on the observed correlation between simulated LC order collapse and the shrinkage of the cell [78]. Such behavior is analogous to an experiment, yet easily extended to understand the effect of long- and short-ranged interaction at a molecular scale, as the small-scale modulation is easily realized through the simulation framework (Figure 7b) [79]. This method was later extended by Moon et al. [21]; they described a smectic LCN, where translational symmetry is addressed by considering the longer-ranged molecular interactions through the coarse-grained molecular model, which is another layer of multi-scaling.

Mehnert et al. [80] also presented another type of constitutive model describing photo-mechanical behavior by decoupling energies into an elastic model and one related to the electric field (i.e., the Maxwell model), with the order parameter coupled to the field. In this model, light is described by a time-averaged electric field. Although this model lacks the statistical description of the polymer distribution found in the VWT energy model, it is able to accurately capture light–solid interactions by incorporating the Maxwell equations into the model.

Furthermore, a micromechanical approach toward spontaneous behavior is presented by Brighenti et al. [19,81]. The statistical changes in the polymeric chains are correlated with the stimuli-induced shift in chain length (Figure 8), which is similar to the descriptive model using molecular dynamics simulation [18] and statistical polymer physics [82]. The resulting micromechanical model was found to predict several spontaneous behaviors of the network induced by external stimuli, e.g., temperature and electric field, by modeling changes in network anisotropy.

Nevertheless, one may easily notice that the aforementioned eigenstrain models lack consistency with the energetic models and the thermomechanical models, as they focus on either reproducing the structural deformation of LCNs or incorporating microscopic aspects of the polymer. Typically, the model assumes that the principal direction of the spontaneous deformation remains coaxial to the original nematic direction. Although the stationarity of the LC orientation is widely observed in LCN experiments, it is still true that the soft behavior cannot be modeled comprehensively as the mechanical–order coupling [57] is ignored.

Motivated by this, there are works that consider bridging these two seemingly different constitutive models. Based on the thermomechanical constitutive model of Jin et al. [57], Lin et al. [63] derived the quasi-soft linearized beam model, which includes both the linearized constitutive equation and the eigenstrain model. The derived eigenstrain is a function of the shape parameter, such as the deformation induced by an order collapse in Equation (2), as well as a director that undergoes an infinitesimally small rotation when shear stress is given. This model is adopted by Chung et al. [83], in which the quasi-soft constitutive model and eigenstrain formulation are used in the two-dimensional plate model.

## 4. Structural Analysis of Nematic Solids

Structures bearing LCNs fully or partially demonstrate several phase behaviors. Understanding these behaviors requires not only knowledge of unique material properties, such as anisotropy and the alignment of the LC molecules affecting the microscopic mechanical and optical behaviors, but also knowledge of the macroscopic configuration of the structure including assumed kinematics, boundary conditions, external fields, and the geometry and topology of the structure, as well as spatial and temporal changes in the constituents considered in Section 3. These models can be used to predict the behavior of LCNs under various conditions, such as different temperatures and external fields, and can be useful in designing and optimizing products made from LCNs.

### 4.1. Soft Elasticity of LCN Structures

The soft behaviors of LCN structures are widely studied, as they exhibit the peculiarities of LC-mechanical coupling and are observed through changing optical texture. Modeling these behaviors requires constitutive models that determine the stress state and mesogenic rotation for a given strain and instantaneous internal variables such as temperature and polymer shapes.

One of the early successes in this area includes the work of Conti and Dolzmann [84], who implemented the quasi-convexation technique into a standard static finite element model. In this model, the deformation gradient at the material point is considered as a rank-one composite of microscopic behaviors following [44]. Based on the relaxation of the non-convex energy [44], the model determines the optimal configuration of rotations (i.e., relative portion and configuration of deformation gradients of which differences are rank-1 matrix [44]) and stress distribution of the bi-clamped LCN strip under uniaxial loading. As shown in Figure 9a, this method addresses the local oscillation of the director as well as the mechanical behavior of the stretched monodomain, as the evolutions of the stress and mesogenic rotation field, which are intertwined, are obtained, and are illustrated with FE modeling in which we can see that the wrinkling in nematic state $\left(r>1\right)$ is decreased, while wrinkling occurs in a pure elastic sheet $\left(r=1\right)$ Figure 9b. This method is particularly interesting, as the effect of the boundary condition was saliently demonstrated near the clamped region.

Although this method is preliminarily based on assumptions about the deformation gradient and therefore not readily applicable to three-dimensional cases, it is still widely employed and has been extended to the different types of transition [85,86] and polymer elasticity [42]. Furthermore, these numerical models provide valuable insights into the tunability of wrinkling [53,87].

In addition, other aspects of soft elasticity are also addressed by the finite element models incorporating variationally consistent constitutive models found in Section 3.2. These include the dynamic behavior of LCNs, e.g., strain rate, hysteresis, and viscosity, as well as the modulation of soft behavior by changing the material parameters such as the nematic alignment and environmental factors such as temperature upon stretch.

Early studies of dynamic LC-mechanical coupling behaviors, such as [55], proposed the coupled governing Equation (6). In contrast to relaxation methods, the work introduced the idea of an element-wise pseudo-director that corresponds to local mesogenic orientation. This assumption greatly alleviates difficulties in modeling the structure, as the nematic directors can be explicitly distributed. As a result, the phase behaviors of LCN structures, both initially monodomain and polydomain, can possibly be investigated.

First, an ideal monodomain structure is assumed to have a uniform mesogenic orientation n and to stretch perpendicular to its direction. As shown in Figure 10a, distribution of the pseudo-director is represented. A domain where directors align horizontally (n in Figure 10a) is colored green, while the one having directors rotated counterclockwise (i.e., positive) within 90 degrees is marked by red color, and otherwise is marked as blue. Without eigenvalue analysis to determine the mixture phase (e.g., smectic, solid, nematic found in Equation (3)), the change in local directions captured through the numerical analysis agrees well with [44], where mesogens first start to rotate where the stress is high. However, it is worth noting that small-wavelength stripe textures are not observed in the model. Second, the polydomain structure is generated by randomly distributing director n throughout the domain, as shown in the left of Figure 10b similarly to Figure 10a. The structure is thus isotropic in the averaged sense and corresponds to the case where the LCN material is crosslinked in a heated environment (i.e., isotropic) by executing a quenching simulation; the aggregation of the pseudo-directors is observed, and an increase of the monodomain region is observed. Such numerical results agree with the fact that, without two-step crosslinking [11], a single crystalline monodomain structure is hard to obtain, thus shedding important light onto LCN behaviors.

Explicit representation of mesogen orientations has been widely adopted in investigating more intricate soft behaviors. Zhang et al. [58] presented simulations employing element-wise orientation into the FE model where the variationally consistent thermomechanical constitutive model and governing equation of Equation (8) are implemented. Variables found in a rotational momentum balance equation, namely the mesogen directors, shear stress, and rigid body rotation, are evaluated at the material point. The study finds that LC reorientation is indeed a rate-dependent process, while stronger rate dependence is observed as the angles between the director and the loading axis increase. Additionally, the appearance of stripe domains and their configurations are found to depend on the aspect ratio and initial director orientation. The finite element model employing a dissipative constitutive model is further exploited in [64,88]. These works demonstrated highly anisotropic loading–unloading behaviors that are dependent on the loading axis different from the nematic alignment direction; morphology generation as well as behavior and dissipation of energy increases as the angle between two directions increases. These simulated behaviors were not only consistent with the experimental observations but also deepened insight on the potential of programming the instability found in the LCN materials.

It is notable that there are types of finite element methods that do not rely on constitutive models. Instead, the equations of motion are built upon Hamiltonian dynamics, and the total free energy is minimized through numerical dissipation. Although similar to the continuum theory of LCN [59], as the relaxation technique is not utilized, these finite element models are easily scaled to multi-dimensional structures. Based on the assumption of small deformation, Mbanga et al. [89] presented total Hamiltonian H as a function of strain $\epsilon^{p}$, velocity $v_{i}$, and the Q-tensor $Q^{p}$:$$H=H_{e}+H_{n}$$(12)$$H_{e}=\sum_{p}\frac{1}{2}\epsilon^{p}:C:\epsilon^{p}V_{p}+\sum_{i}\frac{1}{2}m_{i}v_{i}^{2}$$
$$H_{n}=\sum_{p}V_{p}\left(-\alpha \epsilon^{p}:\left(Q^{p}-Q_{0}^{p}\right)+\beta \left|\left|Q^{p}-Q_{0}^{p}\right|\right|\right)+\gamma \sum_{p\neq q}\left|\left|Q^{p}-Q^{q}\right|\right|$$
where $H_{e}$ and $H_{n}$ refer to the Hamiltonians of an isotropic elastic solid, which is a function of infinitesimal strain ϵ, and the nematic potential, respectively. Superscript p and subscript i denote element- and node-wise variables, and $V_{p}$ is the volume of the element. The first term of $H_{n}$ comprises the terms considering strain–order coupling and the crosslink memory of the order parameters, which are non-typical energies. The second term of the $H_{n}$ penalizes long-range distortion of the energy. The resulting elastodynamics equation updates nodal positions and velocities using explicit multistep time integration. Numerical damping is assumed as the particle-wise force. Although many of the parameters as well as damping coefficients are not strictly defined and the results are rather phenomenological, these models are widely used to describe the behavior of LCNs to describe the effect of strain rate and the angle between tension and nematic director [89], as the governing equation is embarrassingly parallel [90].

### 4.2. Spontaneous Behaviors

The use of constitutive models that account for stimuli responsiveness in finite element methods enables the prediction of spontaneous behaviors exhibited by LCN materials and structures. Valuable insights into mechanical behavior are drawn from the in silico experiments, which are crucial for understanding the structural responses under different conditions. In this short review, we mainly focus on the out-of-plane deformation of thin sheet nematic solids for two reasons. Firstly, in-plane deformations such as uniaxial contraction and swelling behaviors can be relatively straightforwardly predicted via simple eigenvalue models of lower dimension (i.e., 1D), and so we will not discuss FE models when they can be reduced to simple 1D models, although such stimuli responsiveness is still an active area of research. In addition, thin LCN structures are typically fabricated, as such a shape facilitates the control of the nematic orientations. Having a high aspect ratio and low bending rigidity, the structure is prone to deform out-of-plane for several accounts, including eigenstrain of thickness gradient and mechanical frustration, to name a few.

The bending behavior of the LCN strip is first analyzed in [40], where the simple eigenstrain model is introduced to the classical Euler–Bernoulli beam. The eigenstrain is assumed to be continuously changing through the thickness, following a nature of the light as a stimulus and its decay. The eigenstrain therefore varies in the thickness direction and uniform to the axial direction; moreover, by changing the gradient of the eigenstrain, experimentally observed variations of bending behaviors are reproduced. Employing the Rayleigh–Ritz method and an assumed polynomial displacement field with von Karman nonlinearity [91], Dunn [70] represented the bending of the LCN thin shell and curvature evolutions for the given magnitude of stimuli. In his work, the eigenstrain model is extended to a two-dimensional tensor embedded onto the neutral plane as a metric tensor that reads:(13)$$\epsilon^{mono}=\left\{\epsilon^{*},\;\nu \epsilon^{*},\nu \epsilon^{*}\right\};\;\epsilon^{poly}={\left\{\epsilon^{*},0,0\right\}}^{T}$$
where ν denotes the Poisson ratio that applies to the type of eigenstrain and mesogenic configuration (either monodomain or polydomain). As shown in Equations (8) and (10), the principal axis of the eigenstrain strongly influences the behavior of the structure. When the eigenstrain, either $\epsilon^{mono}$ or $\epsilon^{poly}$, is uniform on the domain, simple bending behaviors are confirmed by experiments [70], as shown in Figure 11a. This model shows cases of the enhanced modeling capability of the eigenstrain when it is extended to two-dimensions. The simple analytic model constructed using the Ritz method was later compared with the nonlinear finite element model and found to agree for the moderate magnitude of eigenstrain [71].

As noted in Section 3.3, eigenstrain models are not unique and are dependent on their constitutive models. For instance, the order-thermo-mechanical coupling model suggested in [57] is utilized to study the light-induced deformation of LCN structures bearing active mesogens in [63]. The decaying intensity of the stimulus changes the overall constitutive relation, namely modulus and eigenstrain. The model explains the quasi-soft behavior as well as the phase behavior and is readily implemented in the commercial software ABAQUS. The model is later extended to large-deforming thin sheets [83], where diverse out-of-plane behaviors ranging from bending, twisting, and the evolution of anticlastic curvature are observed using the corotational shell element. The graphs in Figure 11b show that the specimen using light-induced bending has certain characteristics; the thin sheet specimen has more deflection, which was expressed using jet color by experimenting with different thicknesses of the sample. Recently, Brighenti et al. [19] also addressed several aspects of the multiphysical nature found in LCNs by incorporating a micromechanical constitutive model [82] and a dissipative model of nematic transition [58]. The mesoscale descriptions of these mechanical responses were compared to prior experimental observations. The graph shows that the FEM model, which is represented by the blue line, has an agreement with the theoretical model, which is represented by the red line as shown in Figure 11c.

As shown in Figure 11a, the bent surface of the LCN structure can be characterized by a single global Gaussian curvature when uniform eigenstrain is assumed. Conversely, complex out-of-plane deformations can be simulated by varying the eigenstrain spatially and temporarily.

First of all, the deformation of the structure was found to be controlled by changing optical patterns, for instance, by selective crosslinking using polarized light [92]. A complex topography is generated by carefully distributing the eigenstrain and a local metric tensor within the LCN domain. These experimental observations demonstrate the potential of guided spontaneous behavior; a thin sheet of LCN may generate substantial out-of-plane deformation in an abrupt manner (e.g., snap-through) to accommodate mechanical frustration, which is a large blocking force that scales with thickness by finding a director arrangement [72]. This approach differs from designs involving bending, which typically result in small blocking forces proportional to the third power of thickness. The early works studying the incompatible strain and its relation to deformation include [39], where the authors investigated the formation of large changes induced by disclination defects (e.g., +1 defect), which were later correlated with Gaussian curvature and reversibility between flat elastica and non-developable surfaces [93]. The shift between surfaces with different Gaussian curvatures is also studied by [94], where the overall energy cost of the LCN structure is far smaller than conventional materials, as the in-plane stretch energy is minimized via the stress-driven reorientation of the director (i.e., soft modes). Modes et al. [95] also presented that the polygonal shapes caused by the incompatible strain are a result of the interplay between the elasticity, activity, and geometry of the material.

Finite element models regarding complex topography were studied by Chung et al. [96]; inspired by the texture-induced nondevelopable surface [38], the model studied the bifurcation of the structure for the given light-induced bending moments and the in-plane stress resultant in the direction of the LC texture. This work is similar to the out-of-plane deformation found in [12] (Figure 12a), which is modeled by imposing differential strain that is found to mitigate the model complexity of incompatible strain. The bifurcation behaviors were also investigated via the asymptotic numerical method (ANM) by Zhao et al. [97]. In the work, the authors discussed the emergence of thermally induced wrinkles on the surface of a cylinder-shaped LCN structure with different angles of θ, which is an angle that affects the shear strain of a structure, and showed different results of the oblique angle α, the angle between the x-axis and wrinkles (Figure 12b) that alleviates the mechanical frustration in the circumferential direction of the shell depending on θ. The effect of nematic orientation and the geometric configurations of the structure to the wrinkling was thereby revealed.

Temporal nonuniformity of the eigenstrain is also widely studied. For instance, the non-linear photomechanical behavior exhibited by thin-shell LCN was investigated via a population dynamics model by Cheng et al. [98]. This showcases that the experimental observation of photomechanics can be considered from the perspective of eigenstrain, thanks to its versatility. Later, the work was extended by Yun et al. [22], where an ad hoc population dynamics model is replaced by the hybridized model in which ab initio calculation and light penetration model are coupled. A similar idea has been explored to investigate the anomalous self-excited oscillation that LCN exhibits. Li et al. [99] first utilized a linear beam model with the minimalistic eigenstrain model that is assumed to be proportional to the amount of exposure to stimuli, which changes due to the oscillation itself. The model is found to be capable of reproducing vibration behaviors, and later extended to the Timoshenko beam [100] and nonlinear dynamical model [101]. It is also worth noting that the idea of time-dependent eigenstrain is also employed in the self-driven pendulum [9] and pulsating disk [102].

FE models that are not based on the Galerkin method are also presented to understand these intriguing time-dependent behaviors. Zhu et al. [103] considered the Helmholtz free energy of the LCN as well as Rayleigh dissipation, which is solved via an implicit–explicit scheme and the spectral method (Figure 13a). Hamiltonian-based simulation, namely the elasto-nemato-dynamics model [89], is also widely employed to simulate dynamic evolution of complex topography [37,104], thanks to the model’s simplicity and scalability (Figure 13b).

## 5. Discussion and Future Research Avenues

Throughout this work, our aim has been to provide a comprehensive review of continuum models used to describe the phase behaviors of liquid crystal networks (LCNs), which are promising candidates for various engineering applications due to their unique phase behaviors. We discuss two primary phase behaviors: soft elastic behavior and spontaneous deformation, which originate from the rotation of LC molecules due to the statistical change in spatial distributions of crosslinking sites and the change in microscopic polymer shapes corresponding to the LC orders, respectively. We start by presenting a classical model based on the energetics of free energy, where the issue of non-convexity is discussed in depth. The energetic description is followed by constitutive models regarding these phenomena, which vary depending on modeling techniques and fidelities in describing the phase behaviors. Lastly, we describe finite element models that incorporate these constitutive models, emphasizing the importance of these continuum models in predicting the materials’ behavior. Through these models, these continuum-theory-based approaches are shown to have a predictive power to the peculiar behavior of the LCNs and enables deeper understanding of the underlying physics behind the complex interplay.

Nevertheless, there are still limitations in the existing models despite existing endeavors. First, existing constitutive models are limited in considering internal parameters of the LCN, which are determined by manufacturing conditions such as temperature history for each synthesis step, including crosslinking, alignment, and operation, even though these internal parameters substantially affect the phase behaviors. In addition, two important phase behaviors—soft elasticity and spontaneous deformation—are typically considered separately, based on the experimental observation that the LC directors do not reorient significantly because the dominance of one type of phase behavior over the other is typically determined by the crosslink density of an LCN. However, such an assumption is often violated and cannot explain recent experiments regarding chemically different polymeric chains, where the LC order can be disturbed by tension. Furthermore, the eigenstrain found in Equation (9) assumes that the eigenstrain and elastic strain can be additively decomposed, which might not be suitable to describe large-deforming structures. In realistic conditions where the uniaxial stretch is 42%, the formulation should be modified accordingly, e.g., multiplicative decomposition.

In this regard, there are few future research directions, at least from the view of the authors. First of all, a comprehensive multiscale and multiphysics model should be established not only to increase the model accuracy but also facilitate the material’s design by understanding the complex process–property relationship of this novel material. For this aim, a novel material modeling technique that comprehensively requires a data-driven modeling approach combined with theoretical analysis, as well as advances in experimental techniques, would be necessary. In the same vein, a consistent design method that can encompass the multiscale properties of LCN structures altered during the actual fabrication process, including LC texture generations via polarized light and 4D printing, should be established. Lastly, the model can be improved by considering multiphysics, which can also be verified through experiments [105]. We hope that this review inspires further research in this exciting and rapidly evolving field.

## Figures and Tables

**Figure 1 polymers-15-01904-f001:**
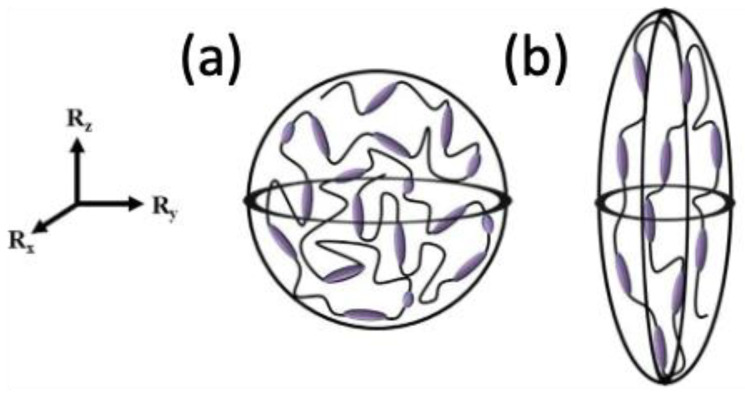
Microscopic description of polymer conformations: (**a**) spherical shape with isotropic LC phase and (**b**) prolate shape with nematic LC phase that is oriented with the director vector $R_{z}$ (Adapted with permission from Ref. [4]. 2018, Sabina W. Ula et al.).

**Figure 2 polymers-15-01904-f002:**
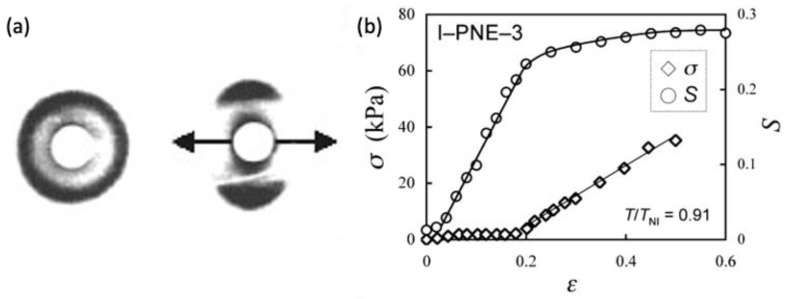
Polydomain–monodomain transition of LCN induced by stretch (**a**) X-ray scattering patterns of the polydomain (**left**) and monodomain (**right**) texture (Adapted with permission from Ref. [31]. 1998, S. M. Clarke); (**b**) stress–strain and order parameter–strain relationships in polydomain nematic elastomers (PNE), illustrating semi-soft elasticity behavior and the correlation between stress, order parameter (S), and strain (ϵ). (Adapted with permission from Ref. [30]. 2009, Kenji Urayama et al.).

**Figure 3 polymers-15-01904-f003:**
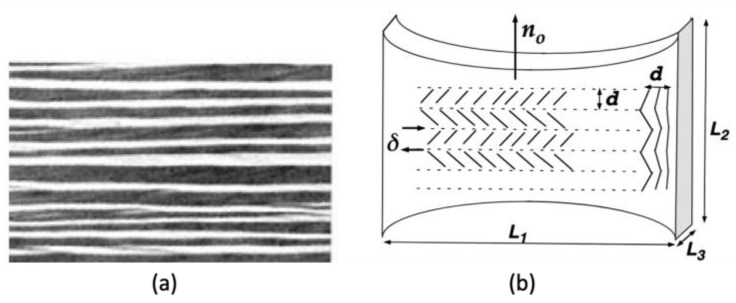
Stretch-induced rotation of monodomain LCN: (**a**) the stripe domain texture; (**b**) the scheme of experimental geometry where stretch is imposed to the direction perpendicular to $n_{0}$ (Adapted with permission from Ref. [29]. 1996, Kenji Urayama et al.).

**Figure 4 polymers-15-01904-f004:**
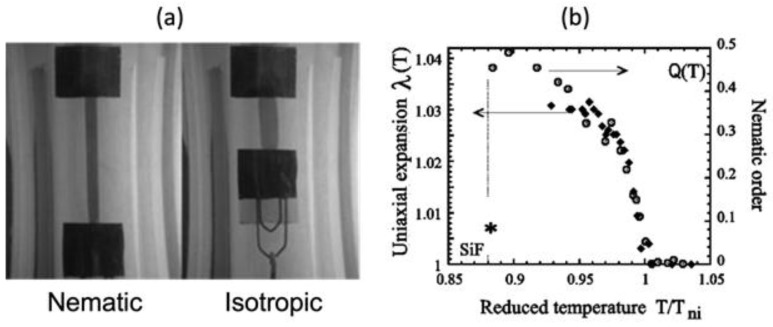
Spontaneous shrinkage of one-dimensional LCN strip: (**a**) strain induced by heating (Adapted with permission from Ref. [35]. 2010, Antoni Sanchez et al.); (**b**) order–strain relationship of LCN with flexible siloxane chain (SiF*) reported in [36]. The star and circle markers denote the uniaxial length change, and change of LC orientational symmetry, respectively (Adapted with permission from Ref. [36]. 2001, S. M. Clarke et al.).

**Figure 5 polymers-15-01904-f005:**
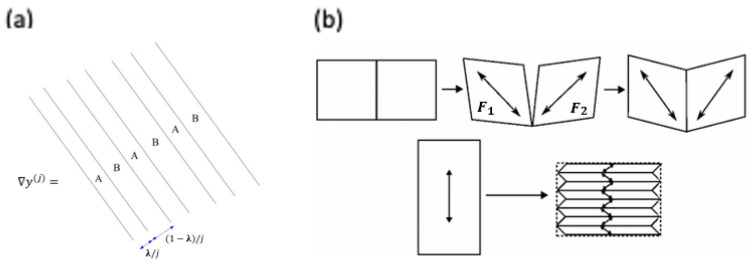
Idea of rank-one lamination of the deformation gradient. (**a**) Example of laminate of compatible gradients A and B with volume fractions λ, 1−λ, respectively (Adapted with permission with permission from [47]. 2004, Ball, J.M). (**b**) Upper: demonstration of material continuity between dissimilar deformations $F_{1},\;F_{2}$ by rigid body rotation. Bottom: lamination leading to stripe domain shown in nematic monodomains (Adapted with permission from [49]. 2009, Biggins, J.S. et al.).

**Figure 6 polymers-15-01904-f006:**
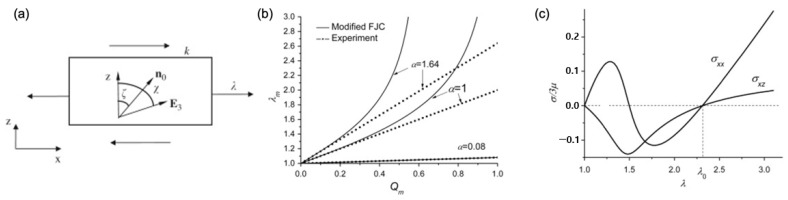
Phase behaviors predicted by Jin et al. (**a**) Uniaxial stretch λ of a LCN material of nematic director $E_{3}$; (**b**) spontaneous deformation $\lambda_{m}$ induced by LC order collapse $Q_{m}$; (**c**) the non-monotonic correlation between normalized stress (σ/3μ ), in both stretched ($\sigma_{xx}$ ) and transverse ($\sigma_{xz}$ ) directions, and the given stretch (λ) observed during the uniaxial stretch when ζ=15° (Adapted with permission from Ref. [57]. 2010, Jin, L.).

**Figure 7 polymers-15-01904-f007:**
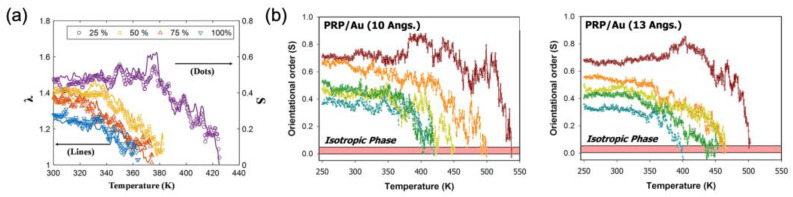
Heat-induced order collapse and the phase behavior of LCNs observed through molecular dynamics simulations: (**a**) correlations of the orientational order S and uniaxial shrinkage λ for the given temperature that are modulated via changing isomerization percentage (Adapted permission from Ref. [23]. 2016, Chung, H. et al.); (**b**) effect of incorporated metallic (Au) nanoparticle to the order collapse, and its dependence on the size of the particle (Adapted with permission from Ref. [79]. 2016, Choi, J. et al.).

**Figure 8 polymers-15-01904-f008:**
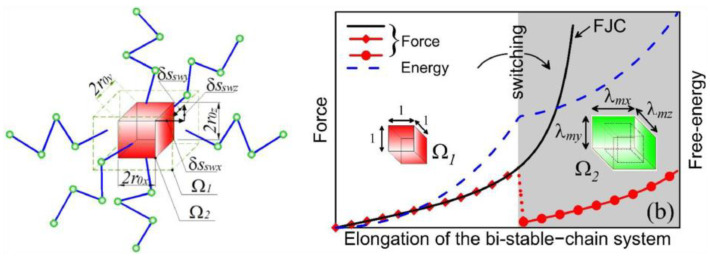
Micromechanical model of polymer network incorporating phase-changing molecules (Adapted with permission from [81]. 2019, Brighenti, R. et al.).

**Figure 9 polymers-15-01904-f009:**
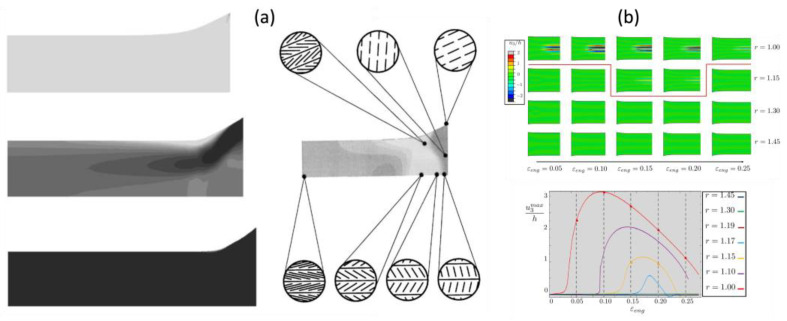
Tension-induced instability of the nematic solid: (**a**) uniaxial tension-induced evolution of the von Mises stress (**left**) and texture (**right**), where darker colors denote higher stress concentration and higher index number (Adapted with permission from Ref. [84]. 2002, Conti, S. et al.) (**b**) under various order parameters (r); changes in the scaled out-of-plane displacement ($u_{3}^{max}/h$) occur with increased stretch ($\epsilon_{eng})$. The higher the order parameters, the fewer micro-wrinkles observed (Adapted with permission from Ref. [53]. 2017, Plucinsky, P. et al.).

**Figure 10 polymers-15-01904-f010:**
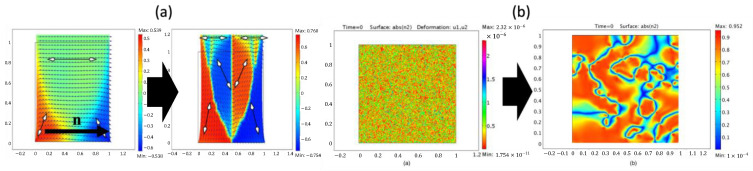
Dynamic texture evolution represented by pseudo-director with colorbar and arrow model, which are expressed according to the local direction. (**a**) Monodomain to stripe texture due to tension; (**b**) quenching simulation of heated LCN (**left**) to cooled LCN (**right**) (Adapted with permission from Ref. [55]. 1992, Oates, W. S et al.).

**Figure 11 polymers-15-01904-f011:**
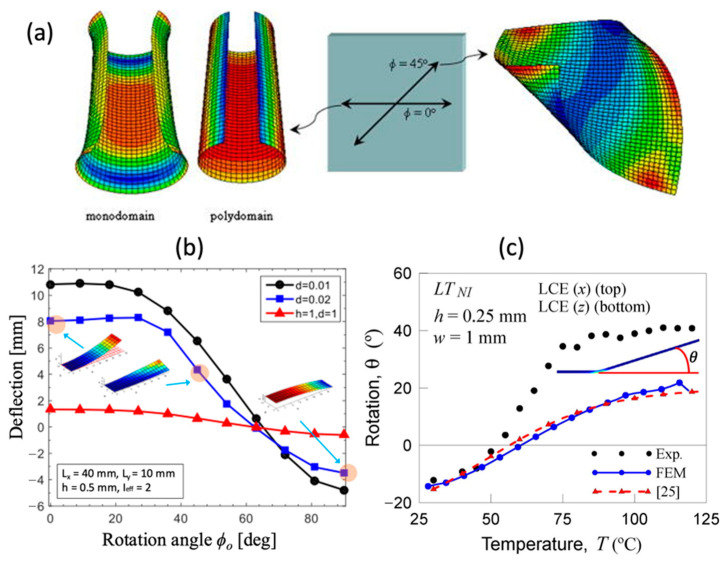
Eigenstrain-based modeling of bending deformation: (**a**) Gaussian curvatures depending on the LC states (Adapted with permission from Ref. [70]. 1992, Dunn. M. L.); (**b**) prediction of light-induced bending (Adapted with permission from Ref. [83]. 2015, Chung, H. et al.) considering photobleaching effect depending on different thickness of element; (**c**) numerical model of temperature-induced bending of LCN numerical and comparison with the experiment (Adapted with permission from Ref. [19]. 2021, Brighenti, R et al.).

**Figure 12 polymers-15-01904-f012:**
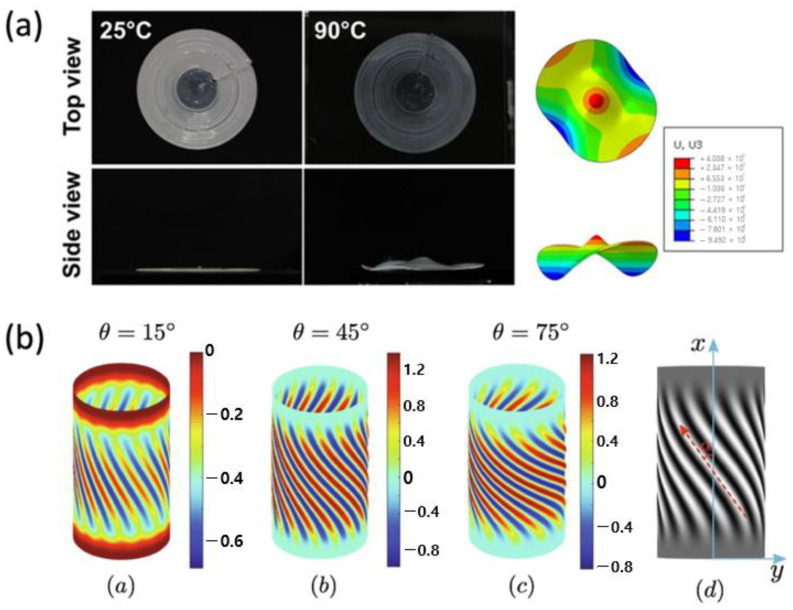
Structural instability observed in (**a**) 3D-printed LCN disk (Adapted with permission from Ref. [12]. 2020, Wang, Z et al.) and (**b**) oblique angle (α) oriented LCN cylinder depending on different θ, which affects shear stress (Adapted with permission from Ref. [97]. 2021, Zhao, S et al.).

**Figure 13 polymers-15-01904-f013:**
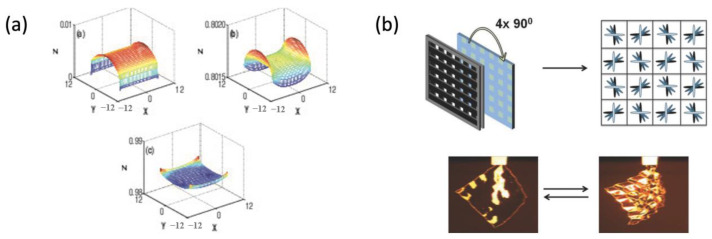
Examples of non-Galerkin-based FE model: (**a**) dynamics of stimuli-induced shape change (Adapted with permission from Ref. [103]. 2021, Zhu, W. et al.), (**b**) evolution of topography analyzed based on nemato-elasto-dynamics model. With the upper 1×4 patterns (red) 90-degree counterclockwise rotated 3 times, the LC films for the test are generated (4×4) (Adapted with permission from Ref. [37]. 2021, Haan, L.T. et al.).

## Data Availability

The data presented in this study are available on request from the corresponding author.

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
