# Peer review of "Recent Trends in Continuum Modeling of Liquid Crystal Networks: A Mini-Review"

_polymers, 2023, doi:10.3390/polym15081904_

Round 1
Reviewer 1 Report
The review by Chung et. al. reports on various continuum models to describe and analyse the mechanical behaviour of different liquid crystal networks, focusing on the soft-elastic response and spontaneous thermomechanical deformations. The models are described in three sections that start from the classical free energy description, to the analysis and development of constitutive models and ends with the presentation of finite element analysis based on these models. The models are summarized and their performance evaluated and compared. The authors conclude the review with a discussion and propose future research directions.
The authors have written an extensive and thorough review that includes relevant and up to date research on the topic. The main text is easy to follow and the progression from the basic free energy description to the developed FE models makes it interesting to read. While readers dealing with LCNs and their modelling will understand the reviewed research with relative ease, young researchers or those who are new to the field, for which these kind of articles are mostly written, will have a harder time to associate the described models with the presented experimental work. This is due to the vaguely described figures and experimental examples given in the manuscript, which are written in such a way, as if it is presumed that every reader is completely familiar with all the work emphasized in the review. The chosen figures are often baffling and lack proper description which would put the theoretical modelling into perspective. Therefore, I cannot recommend this article for submission in the current state, but I would reconsider if the authors would address these issues.
More specifically, I had a hard time understanding what is being investigated in the presented figures. The reader is forced to find on its own what the most of the data means. Please provide the description of the values and curves. One can only guess, for instance, in Figure 6c, what the σ values in the graph mean without consulting the cited article to be sure, or in Figure 7b, it is not clear if this is experimental or modelling data and what exactly is the difference between the two graphs. Here are some more similar issues:
1. Please provide some insight into the materials being investigated and presented (nematic, side-chain, main-chain, azo?) beyond the regular ‘LCN’
2. Figure 8b is not mentioned in the main text at all.
3. How can on distinguish from Figure 9b that the model and experimental data agrees, as stated in the sentence starting on line 669? What does the graph represent?
4. What are the different colours in figure 10?
5. From the presented pictures of the FE model in Figure 11a, it is not clear at all how the FE modelling agrees with experimental data as stated in the manuscript text. The curves in the rest of the graphs are not explained. Please put into proper context.
6. Figure 12- please provide explanation on the written parameters. One can again only presume that theta equals the angle between the cylinder long axis and the wrinkles or the associated nematic director.
7. What does the schematic in Figure 13b means. What is 4x 90°?
There are also some minor issues:
1. The meaning of abbreviations FE and VWT are not properly stated.
2. Some figures are hard to read due to the small size or low resolution.
3. As a suggestion, the last paragraph starting on line 280, page 7, explaining the physical reasons for the collapse of LC order, could be moved to the beginning of the section 2.2, in order to better envision and understand the theoretical description of the spontaneous shape change.
4. Line 404 on page 10 – name Côpič does not match with the citation [60] author’s name Čopič.
5. Citation 48 contains unwanted characters.
Author Response
The authors genuinely express gratitude for the meticulous and thoughtful guidance. With the reviewer's comments, we have observed a significant improvement in the manuscript. We have attentively considered and addressed all the questions. Please see the attachment.

Reviewer 2 Report
This is a well-written and useful review and I recommend publication. I have only a few minor comments that the authors should address before publication.
1) A few lines after Fig 1 the authors use the adjective "uniaxial". I think they should add "prolate", perhaps in brackets, since they use that term later.
2) Why are there quotation marks on line 126?
3) It would be helpful to say something about the curves in Fig 2(b), in the figure caption. This would help the reader who could then avoid hunting for an explanation in the text.
4) I would like to see less use of acronyms. Acronyms are a nuisance for people who are not directly in the field and their avoidance would make the review more approachable and easier to read.
Author Response
The authors appreciate the constructive comments raised by the reviewer. With the suggestions applied, the manuscript has been greatly enhanced. We have thoroughly considered and carefully addressed all the questions. Please see the attachment.

Reviewer 3 Report
The authors provide a complex presentation of continuum theory based models for liquid crystal network. The overall opinion about the manuscript is good as it presentation is quite clear for such a complex topic so it could be considered for publication after a minor revision.
First of all, the authors did not discuss the previous reviews in the field and did not indicate the neccesity of providing a new review with the recent trend in continuum theory modelling. Besides, it is hard to claim a recent trend when about 25% of the references are more than 20 years old. They are classical indeed but not recent.
The second aspect is a technical one: the pictures are too large. Besides, the boundary conditions are not clear from Fig 6a as it is mentioned in the text. A short description might help. In Fig. 7b there is a set of plots without any description.
Author Response
The authors are grateful for the thorough examination. With the comments of the reviewer, the quality of the manuscript has been greatly improved. We have considered and addressed all the questions very carefully. Please see the attachment.

Round 2
Reviewer 1 Report
The authors corrected the paper according to the suggestions and remarks. The review reads very nicely now and all the cited data is understandable. No further revision or improvement is necessary. I suggest the paper for publication in its present form.